# Anatomical Responses of Two Species under Controlled Water Restriction

**DOI:** 10.3390/plants13192812

**Published:** 2024-10-08

**Authors:** Karen Peña-Rojas, Sergio Donoso, Carolain Badaracco, Paulette I. Naulin, Bárbara Gotor, Alejandro Riquelme

**Affiliations:** 1Mediterranean Forests Laboratory, Faculty of Forestry and Nature Conservation, Chile University, Avenida Santa Rosa 11315, La Pintana 8820808, Santiago, Chile; kpena@uchile.cl (K.P.-R.); sedonoso@uchile.cl (S.D.); alriquel@uchile.cl (A.R.); 2Plant Biology Laboratory, Faculty of Forestry and Nature Conservation, Chile University, Avenida Santa Rosa 11315, La Pintana 8820808, Santiago, Chile; pnaulin@uchile.cl

**Keywords:** water restriction, *Cryptocarya alba*, *Quillaja saponaria*, leaf anatomy

## Abstract

Quillay (*Quillaja saponaria* Molina) and peumo (*Cryptocarya alba* [Molina] Looser) are two tree species endemic to Chile that grow in Mediterranean climate zones, characterized by a summer season with high temperatures, high solar radiation, and low soil water availability. A study was conducted with 2-year-old *Q. saponaria* and *C. alba* plants and two substrate water conditions: well-watered and controlled water restriction. At the end of the study, anatomical leaf modifications were analyzed. The tissues were anatomically described in transverse sections of juvenile and adult leaves, measuring leaf thickness, cuticle thickness, and cell density of the mesophyll parenchymal tissues. In the young leaves of *Q. saponaria* plants undergoing water restriction treatment, an increase in cuticle and leaf thickness and a decrease in the density of the palisade and spongy parenchyma were observed. In contrast, a significant reduction in leaf thickness was observed in adult leaves of both species with water restriction treatment. The anatomical changes in the leaves of *Q. saponaria* and *C. alba* suggest an adaptation to adverse environmental conditions, such as water restriction.

## 1. Introduction

Leaves are structures that are susceptible to environmental changes, and their anatomy varies with aridity and water availability [1]. In response to a water deficit, plants adapted to Mediterranean environments have structures that allow them to endure dry periods, such as sclerophyllous leaves with thick cell walls, small intercellular spaces, abundant sclerenchyma, and a thick cuticle. In response to drought, plants may modify these structures, reducing leaf surface area and increasing mesophyll thickness, adjusting stomatal conductance, hydraulic conductivity, and photosynthetic rate, which affects leaf and plant growth [2,3,4,5] Tissues with higher water concentrations, such as the mesophyll in leaves, are most affected by drought stress. A decline in the function and structure of these tissues has a negative effect on the photosynthetic rate, which subsequently results in a reduction in energy generation for the entire organism [1,6].

Quillay (*Quillaja saponaria* Molina) is a tree endemic to Chile, distributed from the Coquimbo Region (30° S latitude) to the Araucanía Region (38° S latitude) in coastal, central, and Andean climate zones, from 15 to 2000 m above sea level [7]. It is an important component of the mixed sclerophyllous forest, occupying the upper stratum in mid- and high-altitude areas, in dry and warm or moist and shaded exposures of the Mediterranean climate region [7,8] This species develops in a Mediterranean-type climate, which is characterized by low soil water availability, high solar radiation, and high temperatures during the summer [9]. The species exhibits evergreen, simple, alternate, leathery, elliptical, glabrous, and shiny leaves, which are yellow–green in color and have entirely or slightly denticulate margins [7]. Its anatomical type is characterized by a double layer of cells in the epidermis [10] and a high-density mesophyll [11]. Changes in the anatomy, morphology, and physiology of *Q. saponaria* plants were observed when a water deficit was present. These changes included a reduction in leaf size and surface area, an increase in cutinization, hairiness, rib density, and the thickness of both palisade tissue and whole leaves [10]. Stomatal opening and closing are affected, reducing photosynthesis and synthesis surface [10]. Another observed change was a reduction in leaf and flower expansion. The effects of water deficit on leaf biomass, including the shedding of adult leaves, contribute to water savings, allowing the reallocation of stored nutrients to roots and younger leaves [4,12,13,14].

Peumo (*Cryptocarya alba* [Molina] Looser), like *Q. saponaria*, is a tree species endemic to Chile, distributed between the Coquimbo and Araucanía Regions. It is found especially in the Coastal Range and the Andes Mountains, where it grows at an altitude of up to 1000 m [15]. This species is found in regions with a Mediterranean climate, including marine, cold, and temperate climates, where the minimum temperature ranges from −3.2 °C to 9.4 °C, and the maximum temperature ranges from 16.5 °C to 31.3 °C. Rainfall across the species’ distribution range varies considerably, ranging from 104.4 to 2555.2 mm/yr [16]. It is most commonly found in humid ravines and south-facing slopes [15,17]. The species is shade-tolerant, with photosynthetic rates dependent on soil water availability. Consequently, growth is reduced during the summer season [16]. It is characterized by evergreen leaves with ascending branches and dense foliage. These leaves are simple, alternate, sub-opposite, leathery, and aromatic, with entire and wavy margins [15]. Transverse sections of the leaves reveal the presence of oil cavities within the mesophyll parenchyma cells [11]. The mean cuticle thickness is 4.8 µm on the adaxial surface and 1.5 µm on the abaxial surface. Stomata are exclusively located on the undersurface of the leaves [18]. Ref. [19] indicated that both epidermises (adaxial and abaxial) are composed of small, rectangular cells with cutinized walls. The hypodermis, situated beneath the adaxial epidermis, is comprised of two layers of larger cells in direct contact with the palisade parenchyma. The mesophyll is dorsiventral, exhibiting the presence of circular secretory glands and cells containing calcium oxalate crystals in the form of druses. The palisade parenchyma is discontinuously distributed between the supporting tissue bands, composed of one to four rows of elongated cells with a high concentration of chlorophyll. The spongy parenchyma is characterized by the presence of large intercellular spaces.

The ability of organisms to respond to variations in environmental conditions is known as phenotypic plasticity, which refers to the capacity of a genotype to produce different phenotypes under different circumstances, enabling them to adapt and survive [5]. Several studies have evaluated the anatomical response of species [2,5,20], particularly under conditions of water deficit [14,21]. However, such research is still scarce for the most important components of the Chilean Mediterranean forest, which is considered a global priority site for biodiversity conservation [22]. Due to the above, this study aimed to understand the foliar anatomical response of *Q. saponaria* and *C. alba* plants both under and not under water restriction. Based on the above, it is hypothesized that *Quillaja saponaria* and *Cryptocarya alba* will show anatomical modifications in their leaves in response to the decrease in water availability, which will allow them to maintain their water balance and survive under drought conditions.

## 2. Materials and Methods

A controlled water restriction trial was conducted on 2-year-old *Q. Saponaria* and *C. alba* plants at the Antumapu nursery of the Faculty of Forestry and Nature Conservation at the University of Chile, located in the commune of La Pintana, Metropolitan Region, Chile (33°40′ S, 70°38′ W, altitude 605 m a.s.l.). This site was selected for the study because it allowed for the replication of the conditions under which the plants were grown.

### 2.1. Previous Experiment: Water Restriction

Forty plants of each species, 2 years old, all produced from seeds cultivated in the Antumapu nursery, were selected. The plants exhibited similar characteristics in terms of development (diameter at collar height [DCH], height [H], leaf area, and health condition). The specimens were transplanted into nine-liter plastic pots with a substrate mix composed of local soil, perlite, and sand in a 4:3:3 ratio, supplemented with triple superphosphate (1.2 g/L) and urea (0.32 g/L). The potted plants were then placed in an uncovered area within the nursery, exposed to summer climatic conditions, and periodically watered for a month to acclimate to the new conditions. After the acclimatization period, 15 plants per species were randomly selected and constantly well-watered, representing control treatment (T_0_), while the remaining plants were subjected to water restriction (T_1_). At the beginning of the trial, all pots were watered to saturation, leaving them at 100% of the field capacity in the substrate (FCS). The control treatment *Q. saponaria* plants (T_0_) maintained an FCS between 81% and 93%, whereas in the *C. alba* specimens, it fluctuated between 74% and 100% throughout the trial. In contrast, for the water-restricted plants (T_1_), the substrate water content was gradually reduced until reaching an FCS of 30% to 40%, corresponding to a predawn water potential (Ψa) of approximately −3.5 MPa, which was maintained for 8 weeks. For further details, refer to the article by [13].

### 2.2. Collection and Fixation of Leaf Material

Once the water restriction trial concluded, 10 plants of each species (five specimens per treatment) were randomly selected, and leaf material was extracted. For *Q. saponaria*, 10 leaves from the upper third of each selected plant were collected (five young (new) leaves that developed during the 8 weeks of application of the treatments and five adult leaves), of which only three of each were used, keeping the rest as reserve material in case of unforeseen circumstances. For *C. alba*, five adult leaves from the upper third of each plant were collected. No juvenile leaves were extracted because the plants under the restriction treatment did not form new leaves. As with *Q. saponaria*, only three of the five collected leaves were used. Each leaf was cut in the central part to obtain a sample. The samples were dehydrated and embedded in paraffin using an ethanol–xylene series, a method adapted from [23]. Transverse histological sections of the leaves were stained with Safranin/Fast Green and mounted with Eukitt (Sigma-Aldrich, Saint Louis, MO, USA). The preparations were stored in the Plant Biology Laboratory at the University of Chile.

### 2.3. Observations and Anatomical Measurements

Observations were made with a Zeiss trinocular optical microscope. Measurements were taken from microphotographs using a Canon digital camera attached to the microscope, using the UTHSCSA (University of Texas Health Science Center at San Antonio) Image Tool program. The anatomy of the tissues observed in the transverse sections of each leaf was described. The thickness of the leaf, cuticle, and density of the parenchymatous tissues (palisade and spongy) were measured, the latter determined by counting the number of cells of each tissue present in an area of 100 × 100 µm.

### 2.4. Statistical Analysis

The analyses were conducted using a completely randomized design, with 150 sample units per treatment, leaf type, and species. Each leaf section used for observations and measurements was considered a sample unit, and the treatments were the irrigation conditions to which the plants were subjected: control (well-watered; T_0_) and experimental (water restriction; T_1_). The results obtained by leaf type, species, and treatment were analyzed through an analysis of variance (ANOVA) for each type of measurement performed (*p* = 0.05).

## 3. Results

### 3.1. Anatomical Description

#### 3.1.1. *Quillaja saponaria*

In juvenile leaves, a wavy adaxial cuticle was observed in control and water-restricted treatments (T_0_ and T_1_, respectively). In the case of leaves from water-restricted plants (T_1_), this cuticle was well developed on both surfaces, while in leaves from control plants (T_0_), it was only evident on the adaxial surface. In general, the adaxial and abaxial epidermises were composed of a single row of cells, although there were sectors of the adaxial surface where they were biseriate. Stomata were only located on the abaxial epidermis. The mesophyll was dorsiventral in both cases (adaxial and abaxial) and was composed of 4–6 rows of palisade parenchyma cells and 3–5 rows of spongy parenchyma cells. Abundant vascular tissue and drusa-type crystals were found within the leaf mesophyll.

Adult leaves had a wavy adaxial cuticle that was well-developed on both surfaces of the leaf in both water-restricted treatment (Figure 1) and well-watered (Figure 2). The adaxial and abaxial epidermises were composed of a single row of cells, with stomata only found on the abaxial epidermis. In both cases, a dorsiventral mesophyll was observed, composed of 3–5 rows of palisade parenchyma cells and 4–5 rows of spongy parenchyma cells, between which drusa-type crystals were observed. In addition, some trichomes were observed on the adaxial surface of adult leaves.

#### 3.1.2. *Cryptocarya alba*

In the leaves of *Cryptocarya alba* plants, a smooth cuticle was observed, thickened on the adaxial surface and not very evident on the abaxial surface. The adaxial epidermis was formed by 1–2 rows of cells, whereas the abaxial epidermis only had one row of cells, in which the stomata were found inserted.

The mesophyll is dorsiventral, where the palisade parenchyma is composed of 3–4 rows of cells, and the spongy parenchyma is composed of 4–5 rows. In the spongy parenchyma, the presence of regularly arranged columns of sclerenchymatous tissue formed by extensions of the vascular bundle along the leaf was evident. In addition, odoriferous cells and drusa-type crystals were observed in the spongy parenchyma (Figure 3).

The main difference between leaves from T_1_ and T_0_ plants was found in the epidermis. On the one hand, on the adaxial surface, the epidermal cells of the inner row tend to become part of the palisade parenchyma. On the other hand, the cuticle is more evident on the abaxial surface epidermis (Figure 4).

### 3.2. Anatomical Measurements

The analysis of variance (ANOVA) showed that there were significant differences between species (F = 4.55; *p* = 0.0212) but not at the treatment level (F = 2.85; *p* = 0.1045) (Figure 5). In adult peumo leaves and juvenile quillay leaves from the water restriction treatment, an increase of 16.8% and 12.7% in cuticle thickness was observed compared to the control treatment (T_0_). In contrast, adult quillay leaves from T_1_ slightly decreased their thickness by 1.82%.

Regarding leaf thickness, significant differences were observed at the species level (F = 37.87, *p* < 0.0001) and treatment (F = 4.65, *p* = 0.0413) (Figure 6). The juvenile leaves that developed during the experiment in T_1_ plants increased their thickness by 37.87% compared to T_0_ control plants. In T_1_ leaves, there was an elongation of palisade parenchyma cells and an increase in intercellular spaces in the spongy parenchyma. Adult T_1_ leaves showed a significant decrease in leaf thickness (4.1%) compared to T_0_ leaves. An incipient disorganization of the mesophyll, especially in the palisade parenchyma, was observed in leaves from both treatments.

### 3.3. Parenchymal Tissue Density

Analysis of variance indicated that the density of palisade and spongy parenchyma varied significantly among the plant species (F = 114.48, *p* < 0.001; F = 39.71, *p* < 0.001) and among irrigation treatments (F = 7, *p* = 0.0142; F = 5.34, *p* = 0.0298). These results suggest that both the species and the amount of water supplied influenced the internal structure of the leaves.

In juvenile leaves, the mean cell density of the palisade parenchyma and spongy parenchyma of T_1_ plants decreased by 30.19% and 15.32%, respectively, compared to T_0_ plants. This decrease was due to an increase in the size of the cells of both parenchymas and an increase in the intercellular spaces of the spongy parenchyma. Alternatively, in the case of adult leaves, the density of the palisade parenchyma (Figure 7) decreased by 4.04% in T_1_ plants. Nevertheless, the spongy parenchyma (Figure 8) tended to increase in density by 3.02%.

## 4. Discussion

The anatomical response of adult *C. alba* and *Q. saponaria* leaves was similar in water restriction treatment; however, in *Q. saponaria*, the response varied according to the plant’s developmental stage. The leaf thickness decreases in adult leaves and increases in juvenile leaves. Additionally, the cuticle thickness in young T_1_
*Q. saponaria* leaves increases. This observation aligns with previous authors’ descriptions, indicating that cuticle thickening as a response to water deficit occurs to protect the leaf structure from water loss through transpiration and high radiation by increasing reflectance [24,25,26,27,28,29] It has been determined in *Opuntia* spp. (Cactaceae) that cuticular transpiration decreases when the cuticle increases in thickness, and it also contributes to the plant’s energy balance due to a reduction in net radiation and an increase in reflectance [30].

The significant decrease in leaf thickness in adult T_0_ plants, compared to T_1_ plants, is similar to the results obtained by [31,32,33] These authors attributed this response to the reduction and disorganization of the mesophyll as a response to water restriction, which, at high levels of deficit, becomes irreversible, leading to a reduction in the plant’s photosynthetic efficiency. This response could be related to the shedding of older leaves to reduce water loss and reallocate stored nutrients [12,13].

Reference [34] concluded that in cotton plants (Malvaceae) subjected to water stress, cell size is smaller compared to nonstressed plants, and the thickness of the cell wall is greater. The looser the leaf structure, the more easily it will lose water due to a larger transpiration surface. A smaller leaf thickness implies smaller cells, which maintain turgor at low water potential values (Ψa), contributing to the plant’s resistance to water stress.

The increase in thickness observed in developed leaves of juvenile T_0_ plants, compared to T_1_ plants, is consistent with the findings of [24,26], who describe an increase in the palisade parenchyma. According to [35], an increase in mesophyll thickness was observed in stressed (*Phaseolus vulgaris* L.) plants, likely due to the increased length of the palisade tissue cells and the increased number of cell layers in the mesophyll, especially in the spongy tissue. Consequently, there was an increase in specific leaf weight. Specifically, the resistance of the intercellular spaces to gas exchange is directly proportional to the thickness of the mesophyll and inversely proportional to the volumetric fraction of the mesophyll occupied by intercellular spaces.

A similar response to that of young *Q. saponaria* leaves was found in two varieties of water-restricted *Olea europea* L. According to [36], a thicker palisade parenchyma could contain a greater number of CO_2_ fixation sites, whereas a thicker spongy parenchyma could result in easier CO_2_ diffusion, increasing resistance to desiccation [24].

The average cell density of the palisade and spongy parenchyma in the adult leaves of *C. alba* and the juvenile leaves of T_0_
*Q. saponaria* decreased compared to T_1_ plants. This is consistent with the results obtained by [37,38,39], which established that an increase in the internal air volume of the leaf has positive effects on water efficiency. However, this behavior is opposite to that described by [40,41], who determined an increase in mesophyll density in water-restricted *Eucalyptus camaldulensis* Dehnh. and *Olea europea* L. plants, due to the decrease in cell size that favors the maintenance of cell turgor at low water potentials.

In adult *Q. saponaria* leaves, it was observed that the spongy parenchyma tends to increase in density in water-restricted plants. This result is consistent with [42], who found that the leaves of water-restricted *Ipomoea batata* L. developed more spongy parenchyma. Another example occurs in *Aristotelia chilensis* (Molina) Stuntz, a species that exhibits phenotypic plasticity, a mechanism through which plants can respond to environmental heterogeneity with morphological and physiological adjustments [5]. In forest fragments, *A. chilensis* reduces water loss by decreasing its leaf area, thickening its leaves, both leaf epidermises, and increasing the number of spongy parenchyma layers [43]. This plasticity supports the hypothesis that *A. chilensis* could soon become one of the dominant species in forest fragments in the Mediterranean region, capable of adapting to an environment of accelerated changes [43]. A similar case is that of *Atriplex repanda* Phil. or *Polylepis sericea* Wedd., which have adaptation mechanisms to maintain physiological activity in dry conditions, allowing their establishment in an unfavorable environment [24,37].

*Q. saponaria* does not exhibit a clear mechanism for adjusting to water deficit (osmotic or elastic [13]), but it does have an anatomical leaf response that varies according to the developmental stage. Anatomical leaf changes can influence the physical or biochemical characteristics of gas exchange at the mesophyll cell level. Changes in leaf structure, such as thickness, cell dimensions, and the ratio between the surface area of chlorenchyma cell walls and the tissue cross-sectional area, serve the plant as adaptations to hot and dry environments [35].

The leaves of plants subjected to water deficit seem to accelerate their development, given that all the parameters measured in this study for young leaves with water deficit resembled the values obtained in adult leaves.

Water stress reduces leaf area at the onset of senescence or by accelerating its rate [44]. Dry soil conditions can promote early leaf senescence to reduce water requirements and avoid low water potentials that are detrimental to the normal function of the xylem [45]. Sites with prolonged dry periods are characterized by the presence of drought-resistant semi-deciduous plants, which have lower leaf mass per area ratios, shorter lifespans, and higher CO_2_ absorption [46].

Water stress reduces leaf formation, induces early abscission and senescence, and often decreases the total number of leaves [47]. The leaf area in *Quercus ilex* L. and *Phillyrea latifolia* L. decreased in response to simulated drought [48]. In *Acacia* spp., the reduction in leaf area caused by water stress was attributed to decreased leaf initiation, growth rate, and leaf size [49].

Leaves expand to intercept the maximum amount of light and CO_2_ for photosynthesis and to transpire water for cooling. The degree of leaf expansion is determined in part by genetic factors and partly by predominant environmental factors. Leaf area development is an important factor that can affect the plant’s response to water availability. Increased leaf area decreases in response to induced water deficit. Leaf expansion and the development of transpirational surface also decrease drastically. It was believed that this arrested growth, with limited investment in osmolyte production, helped plants achieve high productivity in environments with intermittent drought and irrigation cycles [50]. The leaf expansion process is sensitive to water deficit. This sensitivity is expressed in terms of smaller cells and a reduction in the number of cells produced by leaf meristems [50]. In several plants, leaf area growth decreased in response to water stress and quickly reversed upon stress removal [50].

The anatomical changes in *Q. saponaria* at the leaf level suggest adaptation to water stress conditions. The observed differences in the response of adult leaves may be related to other ecophysiological responses such as leaf biomass loss, shedding of adult leaves, and increased root system growth [13,51]. This would explain why *Q. saponaria* does not significantly modify all the anatomical variables analyzed, decreasing leaf thickness in adult leaves as *C. alba* does.

The ability of both species to adapt to varying degrees of water deficit is relevant to their persistence, given that climate change models have predicted an increase in the frequency and intensity of drought episodes and irregular precipitation in Mediterranean climate zones [9,52,53,54,55]. This information complements physiological studies aimed at determining the effect of water scarcity on the development and productivity of different species [13,21,51]. Advances in this field will enable the development of new techniques for the management and production of *Q. saponaria* and *C. alba* plants, considering their use and adaptation to future drought scenarios in Mediterranean climate regions [54,55].

## 5. Conclusions

The anatomical changes in the leaves of *Q. saponaria* and *C. alba* suggest a response to water-restricted conditions. In the young leaves of water-restricted *Q. saponaria* plants, an increase in cuticle and leaf thickness was observed, along with a decrease in the density of the palisade and spongy parenchyma due to an increase in cell size, resulting in a noticeable rise in intercellular spaces in the spongy parenchyma. The increase in cells in the palisade parenchyma and the greater internal air volume in the leaf are related to more efficient water transportation. Only a significant decrease in leaf thickness was observed in the adult leaves of both species subjected to water restriction. This is related to the reduction and disorganization of the mesophyll, followed by early induction of senescence and leaf fall to reduce water loss and reassign stored nutrients.

## Figures and Tables

**Figure 1 plants-13-02812-f001:**
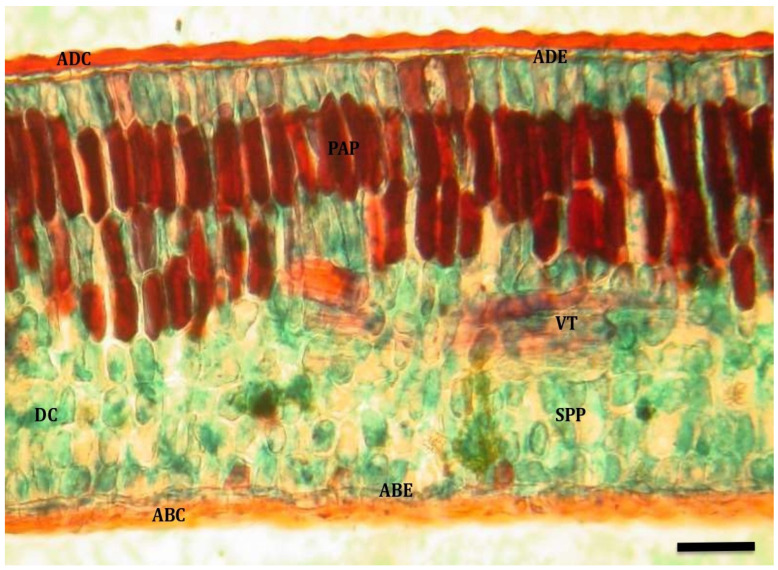
Cross section of adult quillay leaf maintained under water restriction (T_1_; 10×, bar = 40 µm). Reference: adaxial cuticle (ADC); adaxial epidermis (ADE); palisade parenchyma (PAP); spongy parenchyma (SPP); abaxial epidermis (ABE); abaxial cuticle (ABC); vascular tissue (VT); druse crystal (DC).

**Figure 2 plants-13-02812-f002:**
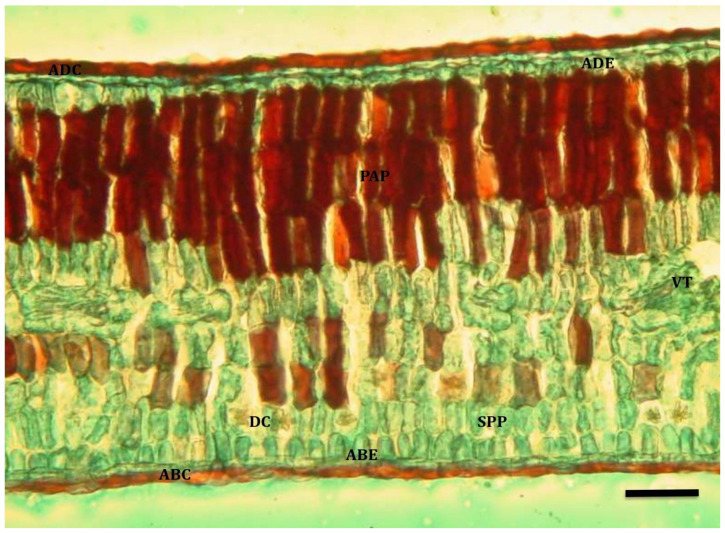
Cross section of adult quillay leaf maintained under permanent irrigation (T_0_; 10×, bar = 40 µm). Reference: adaxial cuticle (ADC); adaxial epidermis (ADE); palisade parenchyma (PAP); spongy parenchyma (SPP); abaxial epidermis (ABE); abaxial cuticle (ABC); vascular tissue (VT); druse crystal (DC).

**Figure 3 plants-13-02812-f003:**
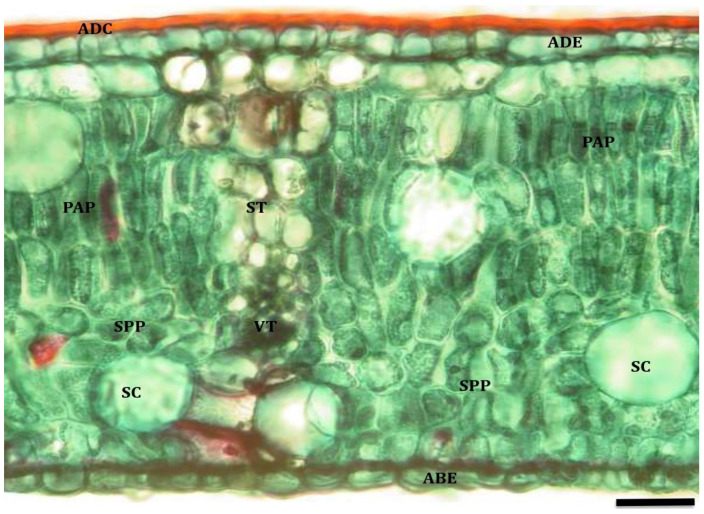
Cross section of peumo leaf maintained under permanent irrigation (T_0_; 10×, bar = 40 µm). References: adaxial cuticle (ADC); adaxial epidermis (ADE); palisade parenchyma (PAP); spongy parenchyma (SPP); abaxial epidermis (ABE); vascular tissue (VT); sclerenchymatous tissue (ST); secretory cells (SC).

**Figure 4 plants-13-02812-f004:**
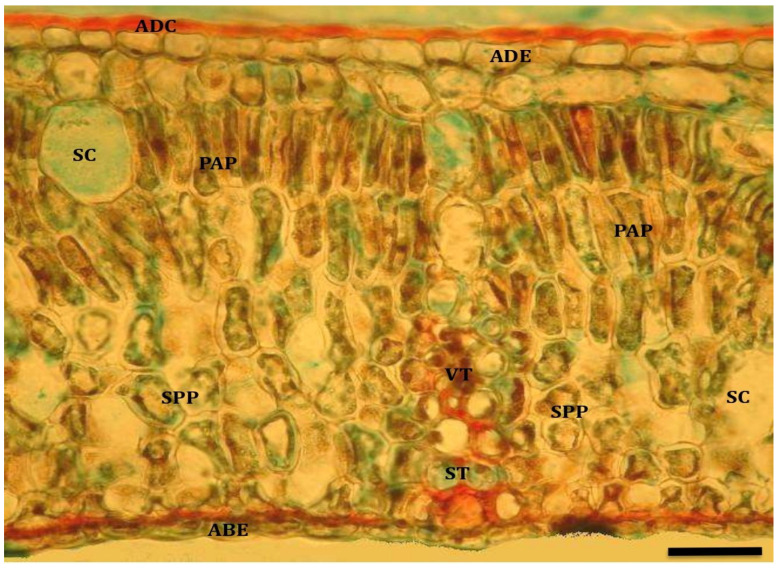
Cross section of peumo leaf subjected to controlled water restriction (T_1_; 10×, bar = 40 µm). References: adaxial cuticle (ADC); adaxial epidermis (ADE); palisade parenchyma (PAP); spongy parenchyma (SPP); abaxial epidermis (ABE); vascular tissue (VT); sclerenchymatous tissue (ST); secretory cells (SC).

**Figure 5 plants-13-02812-f005:**
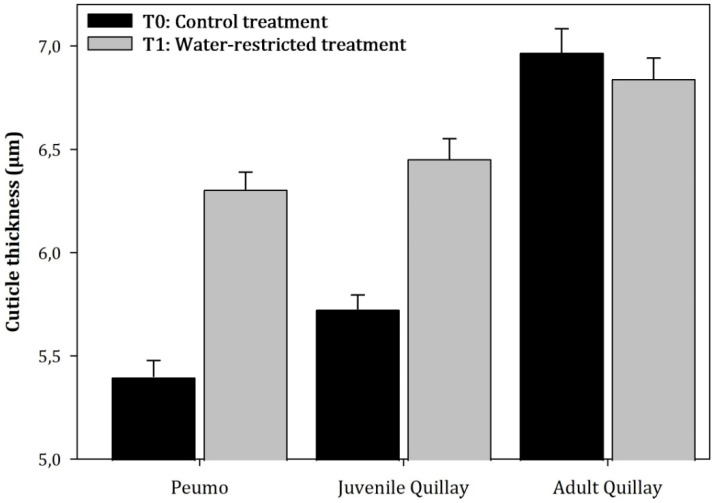
Average value (±Standard error) of the leaf cuticle thickness of peumo and quillay plants subjected to permanent irrigation (T_0_) and water restriction (T_1_; N = 150) (Species: F = 4.55, *p* = 0.0212; Treatments: F = 2.85, *p* = 0.1045).

**Figure 6 plants-13-02812-f006:**
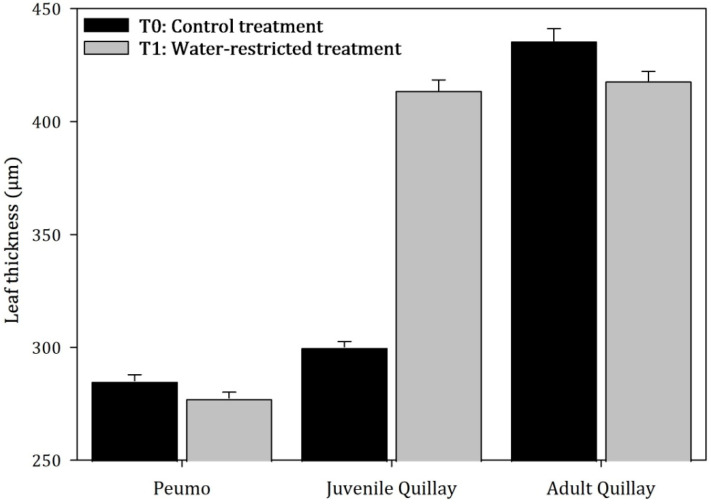
Average value (±Standard error) of the leaf thickness of peumo and quillay plants subjected to permanent irrigation (T_0_) and water restriction (T_1_; N = 150) (Species: F = 37.87, *p* < 0.0001; Treatments: F = 4.65, *p* = 0.0413).

**Figure 7 plants-13-02812-f007:**
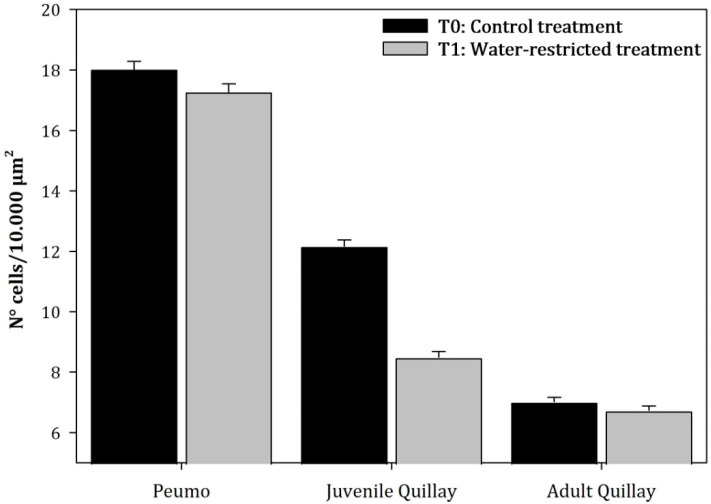
Average value (±Standard error) of the cellular density of the palisade parenchyma of the leaves subjected to permanent irrigation (T_0_) and water restriction (T_1_; N = 150) (Species: F = 114.48, *p* < 0.0001; Treatments: F = 7, *p* = 0.0142).

**Figure 8 plants-13-02812-f008:**
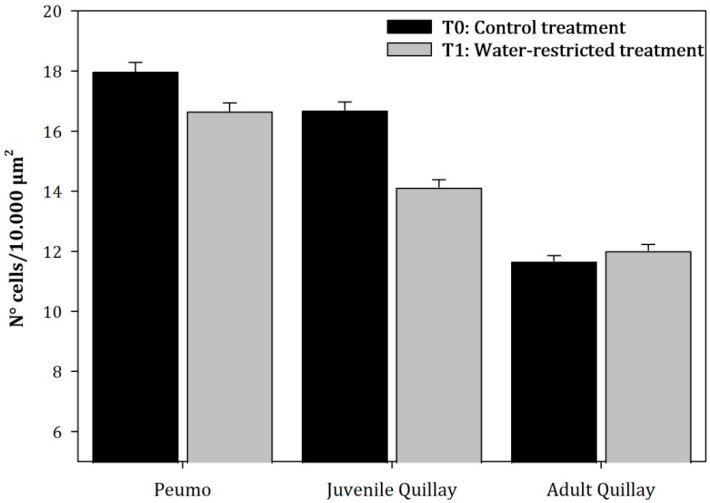
Average value (±Standard error) of the cellular density of the spongy parenchyma of the leaves subjected to permanent irrigation (T_0_) and water restriction (T_1_; N = 150) (Species: F = 39.71, *p* < 0.0001; Treatments: F = 5.34, *p* = 0.0298).

## Data Availability

Data are contained within the article.

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
