# Peer review of "Anatomical Responses of Two Species under Controlled Water Restriction"

_plants, 2024, doi:10.3390/plants13192812_

Round 1

Reviewer 1 Report

Comments and Suggestions for Authors

This manuscript deals with the effect of water stress on leaf anatomy of nursery plants of two tree species. The topic is within the scope of the journal, is interesting and original. As stated in the Introduction section, the topic of this study is something that is well known (phenotypic plasticity due to water stress). However, knowledge about it is still scarce, and it must be expanded to include a good number of species, as well as quantifying its effect based on the degree of water stress suffered by the plants.

The manuscript is well structured and written, although I do not feel qualified to evaluate the quality of English. The bibliography is very up-to-date. The Introduction, Results and Discussion sections are sufficiently developed. However, in the Results section, certain modifications are suggested to make it more complete. The Discussion section is consistent with the data obtained and the parameters analyzed.

Particular comments are highlighted in color in the attached file.

Author Response

All the modifications that were made to the article are highlighted in yellow.

L40 Replace "2,000" by "2000"

Answer: 2,000 is replaced by 2000 (L 40)

L62 Replace "1,000" by "1000"

Answer: 1,000 is replaced by 1000 (L 61)

L66 Remove the separating comma from thousands.

Answer: The comma separating the thousands is removed. "2,555.2" is replaced by "2555.2" (L 65)

L85 It is therefore a well-known process, as reflected in this entire Introduction section. However, it has not been sufficiently studied in all species, nor has the extent of such modifications been quantified.

Answer: While phenotypic plasticity has been extensively researched, there remains a significant dearth of studies examining the microscopic anatomical changes that occur in leaves in response to abiotic stress, particularly water deficit.

L89 The reviewer's comment says "Ok".

L99 The reviewer's comment says "Ok".

L122 Please clearly state that these were new leaves, developed during the 8 weeks of application of the treatments.

Answer: The observation is accepted. It is clarified that the young leaves corresponded to new leaves developed during the 8 weeks of application of the treatments (the modification was highlighted in yellow) (L125-L126).

L168 (FIGURE 1) Please check if it is a juvenile or adult leaf. In the previous paragraph (line 151) it says it is juvenile. (Same comment for figure 2).

Answer: The images in Figures 1 and 2 depict adult leaves. The reference to these figures in the text (L164-L165) has been modified accordingly (change highlighted in yellow).

L168 (FIGURE 1) Please review the magnifications and clarify the meaning. This photograph is hardly 10 times magnified compared to reality. The authors may have used the X10 objective of the microscope, but that does not mean that the image stored on the Canon camera, or the image presented in this article, is 10 times magnified. It would be advisable to include a graphic scale within the figure. (Same comment for figures 2, 3 and 4.)

Answer:  bar equivalent to 40 µm (graphic scale) is incorporated into figures 1 (L172), 2 (L177), 3 (L193) and 4 (L198).

L 168 (Figure 1) Inside the figure, in the abaxial part, replace "ADE" with "ABE".

Answer: In Figure 1, the abbreviation "ADE" is replaced by "ABE" (L172).

L176 It is suggested that, in addition to the percentage increase or decrease, it should be indicated whether the differences between leaves of T0 and T1 were significant after the ANOVA performed, as well as the level of significance obtained (p). Same comment for both species and for both cuticle and leaf thicknesses and cell density (sections 312, 313, 322, 333 and figures 5, 6, 7 and 8).

Answer: The p and F values ​​obtained after performing the ANOVA are added to the text and figures (the changes are highlighted in yellow) (L208-L215, L224-L225, L228-L229, L231-L235, L245-246, L249-L250).

 L177 Please review this value. From figure 5 it looks like it should be 18.2%.

Answer: The text has been corrected and the changes made are highlighted in yellow, since the increase in cuticle thickness for adult peumo leaves and juvenile quillay leaves was 16.8% and 12.7%. The value of 1.82% is correct but corresponds to the decrease in cuticle thickness of adult quillay leaves from T1 (L210-L213).

L183 OK. This phenomenon is common in leaf development. It is well described in the following paragrahp (lines 188-191).

Answer: Yes, this phenomenon is described in more detail in section 3.1.3. called "Parenchymal Tissue Density" (L238-L242).

L201 No es “leafwas” es “leaf was” (modification is highlighted in yellow).

Answer: "leafwas" is corrected to "leaf was" (L189).

L240 Is this referring to the "latter species"? Please clarify.

Answer: Yes, I am referring to the last species mentioned. However, to avoid confusion, "the latter" is replaced by the species name "Q. saponaria" (the change is highlighted in yellow) (L253).

Reviewer 2 Report

Comments and Suggestions for Authors

The manuscript explores the anatomical response of two species under controlled water restriction, the topic is interesting. I have some specific comments for the paper as follow.

1.     I suggest the authors should present why they conduct the experiment of well-watered and controlled water restriction in the abstract briefly.

2.     In the last section of the introduction, the authors lack of the hypothesis of the study, please add it.

3.     At the end of the experiment, how about the survival ratios of transplanted seedlings?

4.     L106, how about the nutrient status of the local soil?

5.     L110, how did the experiment achieve the goal of constantly well-watered?

6.     L120-123, how did the authors distinguish the young leaves and adult leaves?

7.     In the results section, the quantitative analysis is more important than qualitative description.

8.     In all the figures of the manuscript, the statistic analysis and significant difference between CK and treatment group should be presented in the figures, e.g., the P value, et al.

Author Response

All the modifications that were made to the article are highlighted in yellow.

1. I suggest the authors should present why they conduct the experiment of well-watered and controlled water restriction in the abstract briefly.

Answer: The water restriction experiment is described in a limited manner because this information has already been published in another article. In fact, on line 121 it is indicated that for more details it is recommended to consult the article by Donoso et al. (2011). The respective DOI is attached below: http://dx.doi.org/10.4067/S0717-92002011000200009

2. In the last section of the introduction, the authors lack of the hypothesis of the study, please add it.

Answer: The following hypothesis is incorporated at the end of the introduction: "Quillaja saponaria and Cryptocarya alba will show anatomical modifications in their leaves in response to the decrease in water availability, which will allow them to maintain their water balance and survive in drought conditions." (L93-L96).

3. At the end of the experiment, how about the survival ratios of transplanted seedlings?

Answer: Two-year-old plants (L105) were used, therefore, when transplanting them, no specimens died (100% survival).

4. L106, how about the nutrient status of the local soil?

Answer: The substrate used was composed of: 40% local soil (sandy loam texture), 30% perlite, and 30% sand plus fertilizer (L109). More details in the article Donoso et al 2011 (indicated in the article).

5. L110, how did the experiment achieve the goal of constantly well-watered?

Answer: The difference between the weight of dry and wet substrate gives us the substrate's water-holding capacity. Using this information, we can determine the optimal irrigation schedule, which includes both the amount of water to apply and how often. This ensures the substrate maintains the necessary moisture level and allows for a gradual reduction in watering as needed.

6. L120-123, how did the authors distinguish the young leaves and adult leaves?

Answer: Since these are 2-year-old plants, the juvenile and adult leaves can be easily distinguished, since the young leaves have lighter-colored leaves, tend to be rounded and with little prominent leaf venation, while the adult leaves are dark green, leathery and with prominent leaf venation.

7. In the results section, the quantitative analysis is more important than qualitative description.

Respuesta: Se incorporan resultados estadísticos y otros datos cuantitativos para enriquecer la información ya proporcionada.

8. En todas las figuras del manuscrito se debe presentar en las figuras el análisis estadístico y la diferencia significativa entre el CK y el grupo de tratamiento, por ejemplo, el valor de P, et al.

Respuesta: Los valores de p y F obtenidos tras la realización del ANOVA se añaden al texto y a las figuras (los cambios se resaltan en amarillo) (L208-L215, L224-L225, L228-L229, L231-L235, L245-246, L249-L250) y se indicó si había o no diferencias significativas.

Round 2

Reviewer 1 Report

Comments and Suggestions for Authors

The manuscript has been revised and improved taking into account all the comments suggested by the referees.

Author Response

The box is left blank as, at this stage of the process, no additional questions have been identified by the reviewer. We sincerely appreciate the valuable previous comments, which have been essential in improving the quality of our article.

Reviewer 2 Report

Comments and Suggestions for Authors

The manuscript has been improved greatly after the first revision. However, it is not the best way to answer the comments 7 and 8 using their native language but not English.

Author Response

I am attaching the answers to the questions raised again and I apologize for the mistake made earlier.

1. I suggest the authors should present why they conduct the experiment of well-watered and controlled water restriction in the abstract briefly.

Answer: The water restriction experiment is described in a limited manner because this information has already been published in another article. In fact, on line 121 it is indicated that for more details it is recommended to consult the article by Donoso et al. (2011). The respective DOI is attached below: http://dx.doi.org/10.4067/S0717-92002011000200009

2. In the last section of the introduction, the authors lack of the hypothesis of the study, please add it.

Answer: The following hypothesis is incorporated at the end of the introduction: "Quillaja saponaria and Cryptocarya alba will show anatomical modifications in their leaves in response to the decrease in water availability, which will allow them to maintain their water balance and survive in drought conditions." (L93-L96).

3. At the end of the experiment, how about the survival ratios of transplanted seedlings?

Answer: Two-year-old plants (L105) were used, therefore, when transplanting them, no specimens died (100% survival).

4. L106, how about the nutrient status of the local soil?

Answer: The substrate used was composed of: 40% local soil (sandy loam texture), 30% perlite, and 30% sand plus fertilizer (L109). More details in the article Donoso et al 2011 (indicated in the article).

5. L110, how did the experiment achieve the goal of constantly well-watered?

Answer: The difference between the weight of dry and wet substrate gives us the substrate's water-holding capacity. Using this information, we can determine the optimal irrigation schedule, which includes both the amount of water to apply and how often. This ensures the substrate maintains the necessary moisture level and allows for a gradual reduction in watering as needed.

6. L120-123, how did the authors distinguish the young leaves and adult leaves?

Answer:  Since these are 2-year-old plants, the juvenile and adult leaves can be easily distinguished, since the young leaves have lighter-colored leaves, tend to be rounded and with little prominent leaf venation, while the adult leaves are dark green, leathery and with prominent leaf venation.

7. In the results section, the quantitative analysis is more important than qualitative description.

Answer: Statistical results and other quantitative data are incorporated to enrich the information already provided.

8. In all the figures of the manuscript, the statistic analysis and significant difference between CK and treatment group should be presented in the figures, e.g., the Pvalue, et al.

Answer: The p and F values ​​obtained after performing the ANOVA are added to the text and figures (changes are highlighted in yellow) (L208-L215, L224-L225, L228-L229, L231-L235, L245-246, L249-L250) and it was indicated whether or not there were significant differences.